# Paediatric chronic fatigue syndrome patients' and parents' perceptions of recovery

Matthew Robert Harland , Roxanne Morin Parslow, Nina Anderson, Danielle Byrne, Esther Crawley

► Additional material is published online only. To view please visit the journal online (http://dx.doi.org/10.1136/bmjpo-2019-000525).

Centre for Academic Child Health, University of Bristol, Bristol, UK

**Correspondence to**
Professor Esther Crawley;
esther.crawley@bristol.ac.uk

## ABSTRACT

**Objectives** Chronic fatigue syndrome/myalgic encephalomyelitis (CFS/ME) is common in children and adolescents; however, little is known about how we should define recovery. This study aims to explore perceptions of recovery held by paediatric patients with CFS/ME and their parents.

**Methods** Children with CFS/ME and their parents were recruited through a single specialist paediatric CFS/ME service. Data were collected through semistructured interviews with children and parents. The interview questions explored how participants would know if they/their child had recovered from CFS/ME. Thematic analysis was used to identify patterns within the data.

**Results** Twenty-one children with CFS/ME, twenty mothers and two fathers were interviewed. Some children found it hard to define recovery as the illness had become a 'new normal'. Others thought recovery would indicate returning to pre-morbid levels of activity or achieving the same activity level as peers (socialising, education and leisure activities). Increased flexibility in routines and the absence of payback after activities were important. The interviews highlighted the concept of recovery as highly individual with wide variation in symptoms experienced, type and level of activity that would signify recovery. Parents describe how changes in mood and motivation would signify their child's recovery, but children did not reflect on this.

**Conclusion** Some parents and children struggle to define what would constitute complete recovery. However, signs of recovery were more easily identifiable. Definitions of recovery went far beyond symptom reduction and were focused towards rebuilding lives.

### What is known about the subject?

► Little is known about how to define recovery in the paediatric chronic fatigue syndrome/myalgic encephalomyelitis (CFS/ME) population.

### What this study adds?

► Children and their parents struggle to define what would constitute a complete recovery as CFS/ME has become a 'new normal'.
► There is wide variation in definitions of recovery between individuals.
► Recovery definitions go beyond symptom reduction and focus on returning to or achieving the same activity as peers, without payback and with flexibility in routine.

including symptoms, physical activity, social participation (including school) and emotional well-being.[10] Improvements can be therefore be measured in physical, psychosocial functioning and daily activities, making it problematic to define recovery in terms of either symptoms or function.[11 12] Some randomised controlled trials have defined recovery as improvement in multiple domains, others as a return to normal within one domain.[13] However, these decisions on defining recovery have been made by researchers and clinicians and not derived from children and young people with CFS/ME. This paper aims to explore how children with CFS/ME and their parents define recovery from the condition. This is important for future research including treatment trials and outcome studies in paediatric CFS/ME.

## METHOD

### Study design

We recruited participants and their parents/carers to a qualitative study with three research questions: (1) How should we define

## INTRODUCTION

Chronic fatigue syndrome/myalgic encephalomyelitis (CFS/ME) is common in children and adolescents, with a prevalence of 0.6%–2.4%.[1–6] It is characterised by severe fatigue and additional symptoms which get worse on exertion including sleep disturbance, headaches, nausea, musculoskeletal pain, dizziness and cognitive dysfunction.[7] Children with CFS/ME have a reduced quality of life and low school attendance, and many develop anxiety and depression.[8 9]

It is difficult to define recovery in CFS/ME as the condition affects multiple dimensions

recovery? (this paper) (2) Which outcomes are important for children with CFS/ME and their parents? (paper under review) (3) What is the minimally clinically important difference in fatigue and SF-36 physical function scale?[14] Participants and parents/carers were asked about each of these three areas. The answers to our research question on recovery are reported in this paper.

Interviews were used as it was not possible to recruit to focus groups as patients were from a wide geographical area requiring long-distance travel placing a possible burden on children with CFS/ME who experience fatigue and payback. Individual interviews allow more in-depth exploration and interviewing has been used successfully from preschool to high school children with a range of conditions (diabetes, asthma, arthritis and bowel disease[15–17]).

### Participants

Children with CFS/ME and their parents were recruited in outpatient clinics provided by a specialist paediatric CFS/ME service. Children were eligible if they were diagnosed with CFS/ME,[7] mild to moderately affected (not housebound), aged between 12 and 17.99 years, and able to understand the patient information sheets. Maximum variation purposive sampling[18] was undertaken to ensure a range of children (age, gender and disease severity) participated. Children and parents provided the appropriate consent/assent prior to interview. We recruited newly diagnosed patients and patients who were receiving follow-up so that a range of views could be included.

### Data collection

Semistructured interviews were undertaken by three researchers (RMP, NA, DB) between December 2014 and February 2015. Participants were offered interviews at home or on hospital premises to reduce travel.[19] We aimed to interview children and parents separately,[20] but families were given the option of being together if they preferred. Interviews took between 15 and 42 minutes. After 20 minutes, the interviewer checked with the participant that they were happy to continue.

A flexible open-ended interview guide was developed with input from a young person's advisory group within a secondary school (seven female, one male) to ensure the questions were comprehensible and clear to school-aged children. The topic guide included questions on recovery, relevant outcomes to measure and the amount of change considered important (the minimal clinically important difference). This section included questions on how participants knew when they were feeling worse, better and how they would know if they had recovered (online supplementary appendix 1).

Interviews continued until data saturation was achieved.[21]

### Data analysis

All interviews were recorded on an encrypted digital audio recorder and transcribed verbatim. Transcripts were checked for accuracy, anonymised and uploaded to NVIVO V.10[22] to provide an audit trail for data analysis. Thematic analysis[23] was used to identify patterns (themes) within the data in an iterative process. Codes were initially developed 'in vivo' inductively from participants' own language in order to represent concepts important to families with CFS/ME. Codes were then added, merged or split as analysis progressed to reflect emerging dimensions in the data.[24] This allowed higher-level recurring themes to be developed and the data were checked between participants to see if they matched the overall coding. A summary was then written with illustrative quotes to highlight the overall themes and differing perspectives.[25]

### Quality assurance

The first three interview transcripts were reviewed to monitor the proficiency of the interviewers (RMP, NA, DB) in following the topic guide, picking up on cues and avoiding leading questions.[26] The transcripts were then discussed in a meeting with feedback from a senior researcher (EC). The interview transcripts were double coded and themes discussed between researchers (MRH, RMP) to compare coding, highlight any significant differences to improve the trustworthiness of analyses and enhance interpretation.[27–29] Coding was found to be very similar; at times alternative wording was used for the same codes.

### Patient and public involvement

A young person's advisory group comprising secondary school children was involved in the development of the interview topic guide during the design stages of this study.

## RESULTS

### Participants

Twenty-one children with CFS/ME were interviewed, 16 (76%) were female and 5 (24%) were male, mean age was 14.4 years and ages ranged from 12 to 17 years. Seventeen (81%) children had mild CFS/ME and four (19%) had moderate CFS/ME. Twenty mothers and two fathers were interviewed. Most children and parents were interviewed separately, but in 4/21 interviews a parent was present in the room during the child's interview. Five families who had initially expressed interest in participating when being contacted by the researcher did not participate (two were too busy and three children were too ill).

### Difficulty defining recovery

Some children found it difficult to describe how they would know if they had recovered. Living with the condition had become a 'new normal' for them and as they had had CFS/ME for so long, children and their parents could not imagine recovery. Due to the fluctuating nature

of the illness, one older child felt that even if she had recovered she would worry about relapsing:

> I don't really think I would know. I think like now it's constantly at the back of mind like whether or not like it will knock me again… (Child-age-17)

Parents and children both had a tendency to report signs of improvement rather than signs of complete recovery. Some children, but more parents, stated that they would have no difficulty in knowing when they/their child had recovered as they would be 'back-to-normal' engaging in pre-morbid levels of functioning or the same activity as their peers (additional quotes supporting each theme can be found in online supplementary appendix 2):

> I think we would know because she'd be back to how she was before sort of the illness… You know, there's that reference point in time that you can go back to. (Parent)

### Returning to socialising, education and activities
All children reported that recovery would be indicated by a return to activities that were particularly important to them: socialising with friends, attending school/college regularly, and both physical "more swimming and go to the gym" and mental activities "I think I'd just have a bit better memory" (Child-age-15). Often, children described school attendance as important for social not educational reasons. The same categories were also reported as important indicators of recovery by parents.

> Probably school, I'd say—'cause I think, if I get back into school then I can like, get in touch with more people. (Child-age-13)

> Being able to return to activities but do them for longer was important 'walking, erm, walk around for a long amount of time, erm, stand up for a long amount of time'. (Child-age-12)

### Recovery as an individualised process
There was wide variation in types of activities that children and parents described as indicating recovery. For children, these activities included cooking, playing instruments, singing and creating artwork. The types of activities children referred to were relevant for their age. For example, children 16–17 years old talked about wanting to be able to be more independent and 'walk to town', whereas younger children talked about 'sleepovers' and playing with friends, 'run around and play a lot'.

> … and having more independence in that sense. Like most people my age will go out to the shops, or just walk around aimlessly, that seems to be popular,

but like (laughs) I wish I could do that like. (Child-age-16)

> Erm, I miss my sports. Erm, ju- have, have sleepovers, erm, with my friends. (Child-age-12)

For parents, these included activities with the family, travel, holidays and dog-walks. For one family, both the parent and child described increased mobility to be the most significant indicator of recovery as the child was reliant on a wheelchair.

> … knows what she wants to do, if she could do it she would, and her main aim is now to probably try and get round the shops without her wheelchair… (Parent)

In contrast, other parents described much higher levels of activity to signify recovery:

> … on the days when gymnastics was on, she exercised every single day, and that's how I know when I think when she's better it's when she starts physical activity again. (Parent)

The idea that recovery is highly individualised was also evident in the descriptions of symptoms children could tolerate as part of their recovery. Some parents and children stated that no symptoms would be acceptable in recovery. However, some children felt that they would be able to consider themselves recovered even if they still had some symptoms. Children had different views over whether tiredness was bearable or not and which symptoms were tolerable.

> I'd be able to live with my headaches. (Child-age-14)

> … the headaches because they, erm, they just stop her being able to think… (Parent)

### Absence of 'payback'
As well as returning to pre-morbid functioning, children and parents felt that increasing activities should not result in payback (the increase in symptoms following activity) in order for them to be considered recovered.

> I think I'd know if like, I went out and did something, I didn't get payback for it, I think that one, I'd know that I was like really starting to recover. (Child-age-13)

### Freedom and spontaneity
Older children talked about how they had become used to planning their activity in a way that facilitated pacing and reduced payback. They felt that life had become regimented and that they were unable to spontaneously

choose to do an activity/continue without a break. The absence of this rigidity of routine was an important indicator of recovery.

> … like won't do as much during the day so that then I can go out in the evening. Yeah, I think that's like that what I mean by like planning, it's just so I can kind of plan my other activities like accordingly I suppose. (Child-age-17)

> You'd know when she's recovered because she'd be back to normal, because she'd be able to do things, and not just do them but do them without thinking about how much energy she was using. (Parent)

For some parents and children, recovery also incorporated the idea that they would be able to have more flexibility in their sleep routines, without experiencing payback. In contrast, some parents felt that the monitoring of activity would be acceptable in recovery to help limit tiredness.

### Changes in mood and motivation

Parents expressed that recovery would be indicated by a change in their child's mood. For some, this would be a reduction in anxiety, and for others, it would be increased positive mood and more engagement with others:

> Well if she was completely better she would be lively again, bubbly again, joking and laughing again, singing again, but all the time, that's how she would be. (Parent)

Often, parents described that their children would be more motivated when they had recovered. This position was not reflected in the responses of children who often appeared motivated but cautious and limited by their symptoms.

> So if he was better you'd think he'd just be motivated to get up and be physically able. (Parent)

Some parents recognised that changes in personality and preference in activities in their children may be due to factors other than CFS/ME.

> Yeah, she's hit puberty as well so, you know, I think a lot of the sort of withdrawing kind of is a teenage thing as well, like wanting to be on her own. (Parent)

### DISCUSSION

This is the first paper to explore what children with CFS/ME feel about recovery. There is a struggle to define what would constitute complete recovery as CFS/ME has become a 'new normal'. Previous levels of activity prior to becoming ill or keeping up with peers were key markers.

Descriptions of recovery went far beyond symptom reduction and focused on rebuilding lives which incorporated valued activities in multiple areas of life (socialising, education and activities). The ability to be flexible and spontaneous without suffering payback was an important theme. However, the concept of recovery was highly individual; there was heterogeneity in the activities children wanted to return to as well as the extent to which different symptoms would be acceptable in recovery. There are differences between child and parent definitions of recovery in CFS/ME, particularly surrounding the role of mood.

### Strengths and weaknesses

This is a large qualitative study including children (n=21) and parents (n=22) which reached data saturation. We considered we had reached data saturation when no new themes arose from the data.[21] As we only interviewed children who were mild/moderately affected, the results are not generalisable to those who are severely affected or younger than 12 years old. We did not interview children younger than 12 because we did not feel they would be able to answer the questions included in the topic guide. Most children chose to be interviewed without their parents, allowing the exploration of differences between child and parent. In the four interviews where parents were present, the parent was mostly passive, but some contributed a small amount. The child may not have been able to speak freely in these interviews.[30] Children were recruited from one specialist service; therefore, the results may not be generalisable to the whole of the UK. Fewer males were recruited[5] and fathers[2] therefore limiting the findings in these subgroups; however, this is consistent with previous research.[31]

### Results in context with previous literature

Children and parents in this study struggled to define recovery from CFS/ME. This is consistent with the recovery literature in adults with CFS/ME.[32] Trouble defining recovery appeared to relate to uncertainty surrounding self-identity; living with the illness had become a 'new normal'. This reflects the biographical disruption reported in previous research.[33] For a lot of children, the focus of recovery had moved away from symptom reduction and towards re-building their lives and identity; a finding that has been reported in the adult CFS/ME population.[34]

Parents and children varied in their criteria for recovery, corresponding with the variety of definitions used in the literature.[12 32] Parents suggested a return to functioning at pre-morbid levels or at the level of peers would signify recovery. This is consistent with attempts to characterise recovery as being a return to population norms, equivalent to physician-held views.[11] Previous research on adult patient's recovery stories in CFS/ME drew on comparisons between themselves currently and themselves prior to the illness or peers.[35] However, 'pre-morbid' as a marker is problematic as children in this study varied in

their levels of physical function and participation prior to becoming ill, which may change over the course of the illness and memory of pre-morbid levels falls short of an objective measure.[36] Similarly, suggestions that recovery would be signified by an increase in activity levels to match those of the children's peers, reflects the research in the adult CFS/ME population relating to the normalisation of function. However, this is difficult to define for the same reasons as in the adult population (wide variation for different ages and the presence of comorbid health problems).[12]

This study identified key areas of life in which recovery would be recognisable (education, socialising and physical activity). The same areas have been identified in previous research into the development of a patient-reported outcome measure (PROM) for use in this population.[10] However, this paper highlights that the specific details of what recovery means are different for each individual. The ability to be spontaneous without suffering payback was an important marker for recovery in this study; further research is needed to understand how children describe and understand payback.

## IMPLICATIONS

This research highlights the individuality of defining recovery in children with CFS/ME. This suggests that it is vital to involve children in developing any new PROM for paediatric CFS/ME.[20] Use of a patient-generated index in which patients select the most important areas of their lives and rate their health in those areas may be more appropriate.[37] Research studies should consider using questions where patients self-define recovery (eg, 'I have recovered' or 'I have nearly recovered'). Clinicians should be aware of the differences between their own goals for recovery and those of parents and children.

**Acknowledgements** We are grateful to all the children and parents who took part in this study.

**Contributors** EC had the idea for this paper and contributed to writing the paper. MRH and RMP conducted the analyses and wrote the first draft. RMP, NA and DB collected the data, and contributed to the writing of the paper. All authors contributed to the analyses and writing of this paper.

**Funding** This work was supported by a University of Bristol PhD Scholarship. EC was funded by the National Institute for Health Research (Senior Research Fellowship, SRF-2013-06-013).

**Disclaimer** The views expressed in this publication are those of the authors and not necessarily those of the NHS, the National Institute for Health Research or the Department of Health.

**Competing interests** EC was the medical advisor for the Association for Young People with ME (AYME) until 2017.

**Patient consent for publication** Not required.

**Ethics approval** Full ethical approval was obtained from the NRES Committee North West (08/04/2014, ref 14/NW/0170). R&D approval was obtained from the RNHRD (20/06/2014, ref-RBB 427).

**Provenance and peer review** Not commissioned; externally peer reviewed.

**Data availability statement** All data relevant to the study are included in the article or uploaded as online supplementary information.

**ORCID iDs**
Matthew Robert Harland http://orcid.org/0000-0001-6795-9820
Esther Crawley http://orcid.org/0000-0002-2521-0747

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
