## [Reviewer comments · BMJ Paediatrics Open]

ARTICLE DETAILS

TITLE (PROVISIONAL)	PAEDIATRIC CHRONIC FATIGUE SYNDROME PATIENTS' AND PARENTS' PERCEPTIONS OF RECOVERY
AUTHORS	Harland, Matthew; Parslow, Roxanne; Anderson, Nina; Byrne, Danielle; Crawley, Esther

VERSION 1 – REVIEW

REVIEWER	Reviewer name: david vickers Institution and Country: cambridgeshire community services nhs trust UK Competing interests: none
REVIEW RETURNED	02-Jul-2019

GENERAL COMMENTS	This is a very useful qualitative study which reports children's and families views on recovery, and will serve as a useful reminder to clinical teams to individualise their discussion on recovery to take account of patients thoughts, as opposed to having a narrow medical model.
---

REVIEWER	Reviewer name: Jennifer McAnuff Institution and Country: Institute of Health and Society, Newcastle University, UK Competing interests: None
REVIEW RETURNED	06-Aug-2019

GENERAL COMMENTS	Thank you for the opportunity to review this manuscript. This is clearly an important patient population, and I have been pleased to previously review other related manuscripts. However, unfortunately I am unable to recommend publication in BMJ Paediatrics Open. I have four key concerns with the manuscript in its current form: 1. The methods section states that this study is part of a larger qualitative study. However, the authors do not adequately clarify this relationship – what part of the larger study does the submitted manuscript constitute? The wider study is about how to measure recovery, and what improvements are important to patients. The submitted manuscript is concerned with exploring perceptions of recovery. These are very similar aims. Therefore, I am not convinced that this separate manuscript is justified, as I assume the larger qualitative study write-up is in preparation (there is no reference to a paper already published for the larger study, so I assume it's in preparation).2. The introduction is very limited, providing no rationale or justification for the study, e.g. why do 'we' need to define recovery, what would be the advantages to different stakeholders of doing so etc.3. The description of methods is limited, and the account of the
--

	results is fairly superficial. I also note that data collection occurred some time ago, and I would like to understand more about the timeframes of data collection and analysis. 4. The written expression is at times limited, particularly in the second paragraph of the introduction. This makes it more difficult to follow the authors' arguments, e.g. around the context and justification for the study. I hope this feedback is useful to the study team, and wish them well with future submissions of their work.
--	---

VERSION 1 – AUTHOR RESPONSE

Reviewer: 1

This is a very useful qualitative study which reports children's and families views on recovery, and will serve as a useful reminder to clinical teams to individualise their discussion on recovery to take account of patients thoughts, as opposed to having a narrow medical model.

Thank you

Reviewer: 2

Thank you for the opportunity to review this manuscript. This is clearly an important patient population, and I have been pleased to previously review other related manuscripts. However, unfortunately I am unable to recommend publication in BMJ Paediatrics Open. I have four key concerns with the manuscript in its current form:

1. The methods section states that this study is part of a larger qualitative study. However, the authors do not adequately clarify this relationship – what part of the larger study does the submitted manuscript constitute? The wider study is about how to measure recovery, and what improvements are important to patients. The submitted manuscript is concerned with exploring perceptions of recovery. These are very similar aims. Therefore, I am not convinced that this separate manuscript is justified, as I assume the larger qualitative study write-up is in preparation (there is no reference to a paper already published for the larger study, so I assume it's in preparation).

Thank you. We agree that this could be clearer. The wider study had three different aims, and therefore this current paper is unique. We interviewed 21 children and 22 parents to explore the answers to three questions: how should we define recovery (this paper), what outcomes are the most important to measure (under review) and what is the minimally important difference (this paper has now been published [1]).

To clarify this, we have deleted the following in the (methods): "This was part of a larger qualitative study exploring how recovery should be measured in paediatric CFS/ME, and what improvements in fatigue and disability are important to young people and their parents." And replaced this with a new section:

Study Design

We recruited participants and their parents/carers to a qualitative study with three research questions: 1. How should we define recovery (this paper); 2. Which outcomes are important for children with CFS/ME and their parents (under review) and 3. What is the minimally clinically important difference in fatigue and SF-36 physical function scale (13) .

Participants and parents/carers were asked about each of these three areas. The answers to our research question on recovery are reported in this paper.

2. The introduction is very limited, providing no rationale or justification for the study, e.g. why do 'we' need to define recovery, what would be the advantages to different stakeholders of doing so etc.

We have simplified the final paragraph in the introduction to make the flow through more logical.

We have added the following explanation: "Some randomised controlled trials have defined recovery as improvement in multiple domains, others as a return to normal within one domain.(13) However, these decisions on defining recovery have been made by researchers and clinicians and not derived from children and young people with CFS/ME." And we have added the following sentence at the end of the introduction: "This is important for future research including treatment trials and outcome studies in paediatric CFS/ME.

3. The description of methods is limited, and the account of the results is fairly superficial. I also note that data collection occurred some time ago, and I would like to understand more about the timeframes of data collection and analysis.

We state that interviews were conducted between December 2014 and February 2015. We have made some additions to the methods (please see below).

4. The written expression is at times limited, particularly in the second paragraph of the introduction. This makes it more difficult to follow the authors' arguments, e.g. around the context and justification for the study.

We have improved the second paragraph of the introduction by removing duplication and adding the argument for doing this study as discussed above.